# Involvement of *Lophodermella sulcigena* in Endemic Disease of *Pinus mugo* Needles in the Polish Tatra Mountains

Tadeusz Kowalski ⬡, Czesław Bartnik ⬡ and Piotr Bilański *⬡

Department of Forest Ecosystems Protection, University of Agriculture in Krakow, Al. 29 Listopada 46, 31-425 Krakow, Poland; rltkowal@cyf-kr.edu.pl (T.K.); rlbartni@cyf-kr.edu.pl (C.B.)
* Correspondence: piotr.bilanski@urk.edu.pl; Tel.: +48-1266-253-69

**Abstract:** *Pinus mugo* plays a significant ecological role in the natural environment at high altitudes in the mountains including the Alps, Pyrenees, Carpathians, and Balkans. In such severe conditions, it is subjected to the harmful effects of various abiotic and biotic factors. In one of the areas of its natural occurrence in Tatra Mts. (southern Poland), for the last few years, a significant intensification of needle disease has been observed. Symptoms similar to those recorded in Tatra Mts. also occur on other *Pinus* species in Europe and North America, where they are caused by fungi belonging to the genus *Elytroderma*, *Lophodermella*, *Lophophacidium* or *Ploioderma* (Rhytismataceae). The current paper presents the results of research which was mainly aimed at characterization of disease symptoms observed for the first time in Poland on *P. mugo* needles, and identification of the main causal agent with use of the morphological and molecular technique. Based on the analyses performed at different times of the year (2015–2020), it was found that dieback symptoms initially appeared only on first-year needles, a few weeks after their development. Symptoms occur on one or both needles in the bundle. The distal parts of the needles died, while the basal parts remained green. In the following year, mainly in June and July, on the previous year's needles attached to the shoots, mature ascomata can be seen. The fungus *Lophodermella sulcigena* has been identified as the cause of these symptoms. So far, the related species *L. conjuncta* has not been found. The morphological features of the pathogen microstructure produced on *P. mugo* needles are presented. Attention was drawn to certain features that may make its identification difficult, especially in terms of shapes and sizes of ascospores. The phylogenetic position of the identified causal agent in relation to closely related other species was determined. The current results confirmed that *L. sulcigena* shows great phylogenetic similarity to *L. montivaga*, which is found in North America. Nine rDNA barcode sequences of *L. sulcigena* obtained in this work will enrich the NCBI GenBank database. The obtained results, indicating the presence of other fungi in *L. sulcigena* ascomata, which may limit the spread of its ascospores, were also discussed.

**Keywords:** Tatra National Park; dwarf mountain pine; needle disease; fungal microstructures; Rhytismataceae; secondary fungi

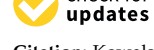



## 1. Introduction

The dwarf mountain pine (*Pinus mugo* Turra) mainly grows as a shrub adapted to various rocky habitats in the high-altitude mountains, including the Alps, Pyrenees, Carpathians, and Balkans [1]. It shows a high capacity for vegetative reproduction and is characterized by a large morphological diversity [2,3]. A related species found in the Pyrenees and Western Alps is distinguished as a separate taxon, *P. uncinata*, which develops a 12–20 m high tree form [2,4]. In the Tatra Mountains (southern Poland), *P. mugo* forms compact groups of shrubs mainly above the montane forest zone, ca. 1.550–1.850 m a.s.l. [3,5].

*P. mugo* plays a significant ecological role in establishing a natural mountain environment. It protects soil from erosion, decreases the avalanche hazard and promotes formation of soil layer. The strong root system of *P. mugo* increases soil pores, thereby improving the hydrological condition of soils [1,2,6,7].

*P. mugo* in severe mountain conditions may be subjected to various types of damage caused by abiotic factors. Stem damage can be caused by rockfall, snow pressure, and wind abrasion [8,9]. Winter drying in dwarf pine clusters can be caused by unfavorable weather conditions. Reduced snow cover exposes plants to increased temperature stress and hinders recovery processes [10]. Reductions in snow cover heights and durations may lead to embolism in *P. mugo*, similar to the seasonal patterns observed in other conifers [11]. At high elevation, drought and frost can co-occur in circumstances where towards the end of winter air temperature increases while soil is still frozen, inducing the so-called winter drought [12]. These factors may lead to the death of needles and shoots and increase of the disease predisposition of plants [11,12].

*P. mugo* also suffers from biotic factors that cause various types of diseases. One of the most important diseases is brown felt blight caused by *Herpotrichia juniperi*, which can lead to destruction of natural regenerations in large areas, especially when thick snow cover persists for a long time [13,14]. In places, similar symptoms are caused by *Allantophomopsis cytisporea* [15]. In some mountain sites the dieback of *P. mugo* is largely due to the fungi of the genus *Armillaria* and *Heterobasidion*, which infect plant roots and cause their rot [16,17]. In turn, the most known diseases of *P. mugo* needles are red band disease caused by *Dothistroma septosporum* and brown spot needle blight caused by *Lecanosticta acicola* [14,18–26]. Among the insects that damage the needles of *P. mugo*, needle shortening gall midge (*Thecodiplosis brachyntera*) can cause significant losses [27].

For several years in Polish Tatra, there has been a high intensity of *P. mugo* needle diseases in some regions [28]. During the preliminary analysis, several different types of disease symptoms on the needles were distinguished, and an attempt was made to determine which fungi can be associated with these symptoms. In total, more than fifty fungal taxa were found in symptomatic and symptomless needles [28]. Three species, *Lophodermium corconticum*, *Lo. pini-mugonis* and *Leptomelanconium allescheri*, have been detected for *P. mugo* in this region for the first time [28,29]. The only fungal species that was found in each symptomatic needle group and also in living symptomless needles was *Lo. conigenum*. Of the biotrophic fungi, only one species has been found: *Coleosporium tussilaginis*. During the in situ analysis, it was found that the most common symptom was the death of the distal parts of this year's needles just a few weeks after their development, while the green basal parts were unscathed. This type of symptom on the needles of various pine species can be caused by fungi of the genus *Elytroderma*, *Lophodermella*, *Lophophacidium* and *Ploioderma* [14,30–33]. In Poland, their occurrence on the needles of *P. mugo* has so far not been recorded [34–36]. Our preliminary study of *P. mugo* needles with this type of symptom showed significant differences between the result of morphological analyses and the result obtained with the use of molecular technique. Due to the great importance of this disease for the reduction of the health status of *P. mugo* in the Tatra Mountains, it was considered justified to conduct more detailed studies. The current paper presents the results of research that had two primary objectives: (i) documentation of disease symptoms observed for the first time in Poland on *P. mugo* needles, and (ii) identification and characterization of the main causal agent on the morphological and molecular basis. An additional goal was to determine the phylogenetic position of identified causal agent in relation to closely related other species. Many rDNA barcode sequences of identified causal agents sent to the NCBI GenBank database can be used for comparison purposes for other researchers. The obtained results should provide a basis for further considerations on the possibilities of protecting *P. mugo* in the high-altitude mountains.

## 2. Materials and Methods

### 2.1. Field Observations

The object of the research was *P. mugo* growing in the Polish Tatra in Chochołowska Valley, where in 2015 a high intensity of the needle disease was noticed, which lasted for the following years. Field observations were carried out on seven plots (1.5–5.5 a) located in dense clumps of *P. mugo* at an altitude of 1556 to 1746 m a.s.l., where an average of

44%–65% needles with discoloration and necrosis were present. The location of these plots and the severity of the needle disease were presented by Bartnik et al. [28]. The samples were collected in autumn 2015 and at least four times in the period from May to November in 2016, 2017, and 2020, and were also used for conducting research on other diseases of *P. mugo* needles [28]. For the purposes of the current study, a detailed description and documentation of selected types of symptoms on this year's needles, and their changes on two-year and three-year-old needles, was performed.

*2.2. Symptom Analysis*

In 2016, a detailed quantitative analysis of the previous year's needles and current year's needles was carried out. For this purpose, 14 twigs in June and 7 twigs in September 2016 were collected, respectively two and one twig, on each of the 7 study plots. The twigs representing the most common disease severity came from the upper part of the crown. They were selected as follows: 10 trees were randomly selected on each plot and the discoloration of needles on the selected twigs was analyzed by assessing its intensity according to the adopted five-point scale. Then, on the basis of the weighted average, the average degree of needle discoloration was determined and the twig that best represented the calculated degree of needle damage was taken for analysis. The twigs were placed separately in plastic bags, transported to the Department's laboratory and placed in a cold store at $-25$ °C. Then, the selected parameters of the needles were measured: the number of needles was calculated depending on the type of shoot (main, lateral), the length of the needles was measured, the degree of discoloration of each needle, the color of the necrotic part and the presence of hysterothecia characteristic of *Lophodermella* species were assessed. Statistica 13.1 [37] software was used for statistical analyses. The Student's *T*-test was used to compare two independent means.

In addition, an analysis of the width of annual rings was carried out in order to determine whether needle disease is accompanied by a decrease in thickness. For this purpose, in September 2020, 35 apical shoots of *P. mugo*, each 50 cm long (7 plots, 5 shoots each), were randomly sampled. In the laboratory, 3 cm thick fragments were cut from the base of these shoots, the cross-sectional surface was sanded and scanned with a resolution of 2400 dpi. Measurements were made with the accuracy of one hundredth of a mm using CooRecoder 7.6 and CDendro 7.6 [38,39]. Thickness increments in a given year were compared in relation to such increments in the previous year.

*2.3. Fungal Isolation and Microscopic Analysis*

To assess the occurrence of hysterothecia of *Lophodermella* type, observations were made using the Zeiss Discovery V12 stereomicroscope (Zeiss, Göttingen, Germany) on needles attached to shoots collected at different times of the year, as well as on needles already fallen from shoots. For microscopic analysis, 1–3 slides from the given fruit body, immersed in water, were prepared. Occasionally, hand-cut sections were made. Great attention was paid to the fact that hysterothecia of various lengths and at various stages of maturity should be included in the microscopic analyses. Measurements of morphological structures were made using the Zeiss Axiophot light microscope with DIC (differential interference contrast) illumination under $1000\times$ enlargement. Numerous cross-sections and analyses of dark spots on needles adjacent to hysterothecia and before their development were also performed. Fungi co-occurring on needles infected by *Lophodermella* and inside hysterothecia were also identified. Among the fungi associated with *Lophodermella*, only *Lophodermium* spp. produced sexual and asexual morphs, and the others only produced asexual stages. For the microscopical analyses, appropriate mycological keys and monographs were used [14,28,31,35,40–45].

Needles with a dead distal part and a living basal part were used for isolation. Surface sterilization was obtained by rinsing the needles in 96% ethanol (1 min), then in a solution of NaOCl (ca. 4% available chlorine, 30 s), 96% ethanol (30 s), sterile water (3 min) and drying in sterilized filter paper. From 50 needles treated in this way, 180 fragments were

excised and placed in the Petri dishes on malt extract agar supplemented with tetracycline to inhibit bacterial growth (MEA: 20 g/L malt extract, 15 g/L agar, Difco, Sparks, MD, USA; 200 mg/L tetracycline, TZF Polfa, Tarchomin, Poland). Petrie dishes were incubated in the dark for at least 10 weeks at 20 °C. Growing colonies were transferred to MEA for new dishes. For the initial identification of *L. sulcigena* cultures, experience obtained during previous research has proven to be useful [46,47].

### 2.4. DNA Extraction, PCR, Sequencing and Phylogenetic Analyses

To verify the morphology-based identification, the nucleotide sequences for four gene fragments of representative cultures were determined: 18S; the internal transcribed spacer regions ITS1 and ITS2, including the 5.8S gene (ITS); 28S region of the ribosomal RNA (rRNA) and translation elongation factor 1-$\alpha$ (TEF1). Genomic DNA extraction, polymerase chain reaction (PCR) amplification and sequencing reactions of the isolates were carried out according to the procedure described in detail by Kowalski et al. [48]. Primers used include NS1 and NS4 [49], ITS1-F [50] and ITS4 [49], LR0R [51] and LR6 [52], and EF1-983F and EF-gr [53]. The resulting ITS and 28S sequences were concatenated because they contained overlapping fragments. All obtained gene fragment sequences in this study were deposited in GenBank with the accession numbers shown in Table 1. The sequences obtained from representative cultures were used as queries in searches using the BLASTn algorithm [54] to retrieve sequences of taxa closely related to them from GenBank (http://www.ncbi.nlm.nih.gov, accessed on 29 December 2022) for phylogenetic analysis. Phylogenetic analyses were performed for concatenated ITS-28S-TEF1 sequences, using methods following Kowalski et al. [48]. The dataset contained all species of the family Rhytismataceae for which three gene sequences were available in the GenBank with *Bisporella discedens* MFLU 18-0691 included as an outgroup following Ekanayaka et al. [55]. The best evolutionary substitution model for the combined datasets was GTR + I + G.

**Table 1.** Fungal isolates obtained in the present study from *Pinus mugo* in Chochołowska Valley (Tatra Mountains).

| Species | Isolate Number [1] | Collection Date | GenBank Accession Number | | |
|---|---|---|---|---|---|
| | | | **18S** | **ITS-28S** | **TEF1** |
| *Lophodermella sulcigena* | Pm349 | 17 May 2016 | OQ288928 | OQ288965 | OQ303894 |
| | Pm350 | 17 May 2016 | OQ288929 | OQ288966 | OQ303895 |
| | Pm402 [1] | 10 November 2015 | OQ288930 | OQ288967 | OQ303896 |
| | Pm422 | 7 July 2016 | OQ288931 | OQ288968 | OQ303897 |
| | Pm423 | 7 July 2016 | OQ288932 | OQ288969 | OQ303898 |
| | Pm426 | 12 July 2016 | OQ288933 | OQ288970 | OQ303899 |
| | Pm427 | 13 September 2016 | OQ288934 | OQ288971 | OQ303900 |
| | Pm640 | 23 September 2017 | OQ288935 | OQ288972 | OQ303901 |
| | Pm641 [1] | 24 September 2017 | OQ288936 | OQ288973 | OQ303902 |

[1] For molecular studies fungal cultures were used, in other cases DNA was extracted directly from the needles.

## 3. Results

### 3.1. Symptoms

*P. mugo* needle disease in Chochołowska Valey in Tatra Mountains persisted throughout the study period, from 2015 to 2020. The disease affected the needles on the main and lateral shoots, mainly in the upper part of the crowns (Figure 1). Based on the analyses performed at different times of the year, it can be determined that the disease had the following course. Dieback symptoms initially appeared only on first-year needles, a few weeks after their development. From July to late summer, dying needles were generally reddish brown in color (Figure 2a). Symptoms occur on one or both needles in the bundle (Figure 2a–c). The distal parts of the needles died, while the basal parts, 0.3–0.8 (1.5) cm long, remained green. Since autumn, numerous needles have become

white-gray (Figure 2b). In the following year, in June and July, ascomata continued to ripen on such previous year needles attached to the shoots (Figure 2c).

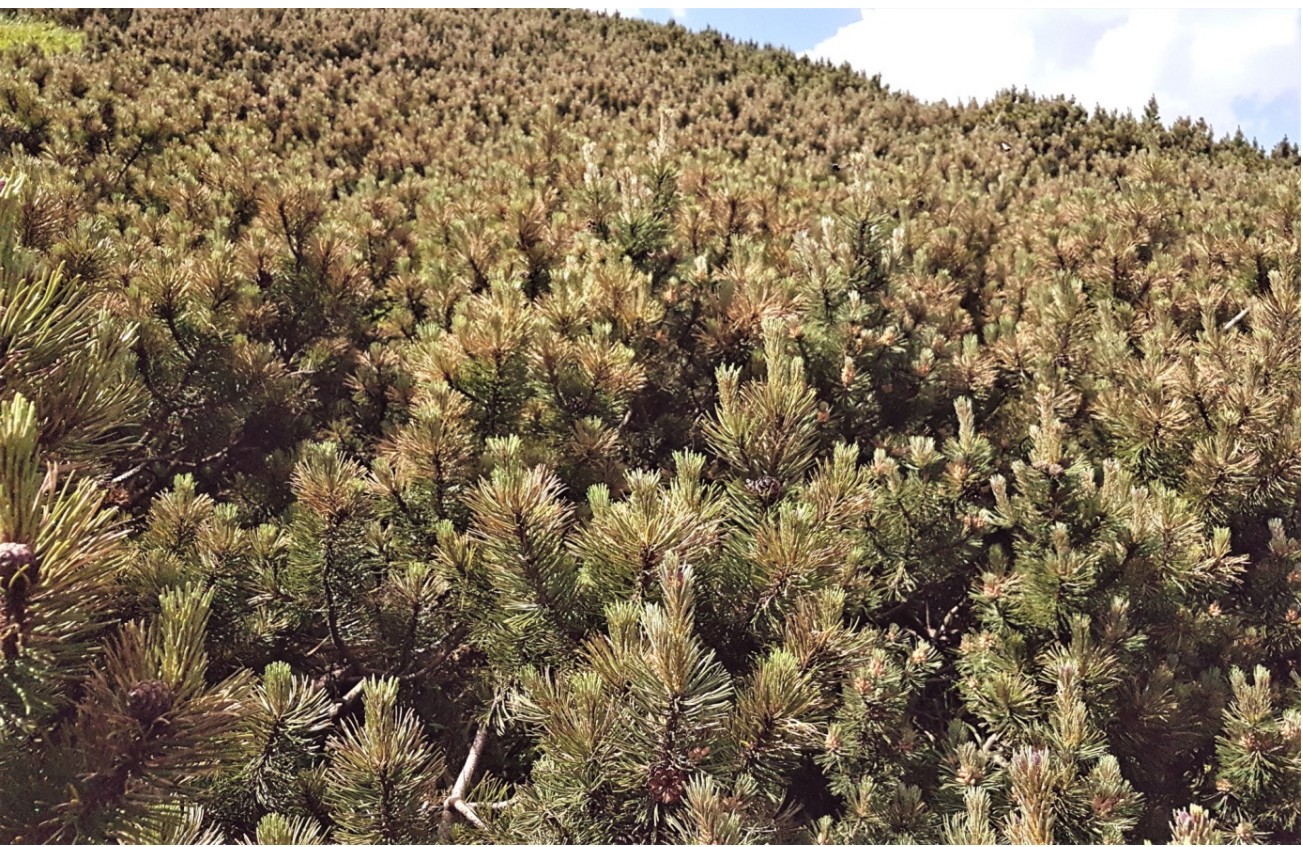

**Figure 1.** *Pinus mugo* in Chochołowska Valley in Tatra Mountains, general view of infested thickets (July 2016).

Other symptoms on some of the needles have also been detected. Transverse red-brown bands were often visible between the green living part and the dead part of the needle (Figure 2d). In turn, dark spots appeared on the dead parts of the needles (Figure 2e–g). They were particularly numerous on the needles of some shoots in the summer of 2016 (Figure 2e). In addition, transverse black stripes were formed on some needles at the periphery of zones with developed ascomata (Figure 2f). Basal parts of previous year's needles began to die off in late summer and autumn which was followed by falling off of these needles.

The characteristics of the needles on the shoots and the severity of some symptoms in 2016 are given in Supplementary Materials Table S1. The density of the needles was higher on the main than on the lateral shoots. In total, needle discoloration was found in 65.8% of the previous year's needles and in 77.8% of current year's needles. In general, symptoms of fungal infection were slightly more common on only one needle in the bundle (Supplementary Materials Table S1). Great variation in the severity of the disease process between individual shoots was found. On the main shoots, the average length of discolored needles was significantly shorter than that of green needles. The type of discoloration covering more than ¾ of the length of the needle occurred in 60.7% of the previous year's needles and in 59.7% of current year's needles. These symptoms were slightly more common on the needles on the lateral shoots than on the main shoots. The vast majority of these needles showed a white-gray discoloration during the analyzed period. Hysterothecia of *L. sulcigena* were present on 41.6% of such needles from last year. In the autumn, the initial stage of such fruitbody could be seen on 3.4% of current year's needles (Supplementary Materials Table S1).

The analyses of the width of annual rings on 35 shoots showed that 42.9% of them had a reduction in growth in 2016 compared to 2015 (Table 2). In the space of the following four years, the share of shoots with decrease in growth compared to the previous year varied from 56.7 to 65.7% (Table 2). The decrease in increment of individual shoots ranged from 2.3% to 72.6%, while the average value of decrease in increment for all shoots between individual years ranged from 20% to 25.8% (Table 2).

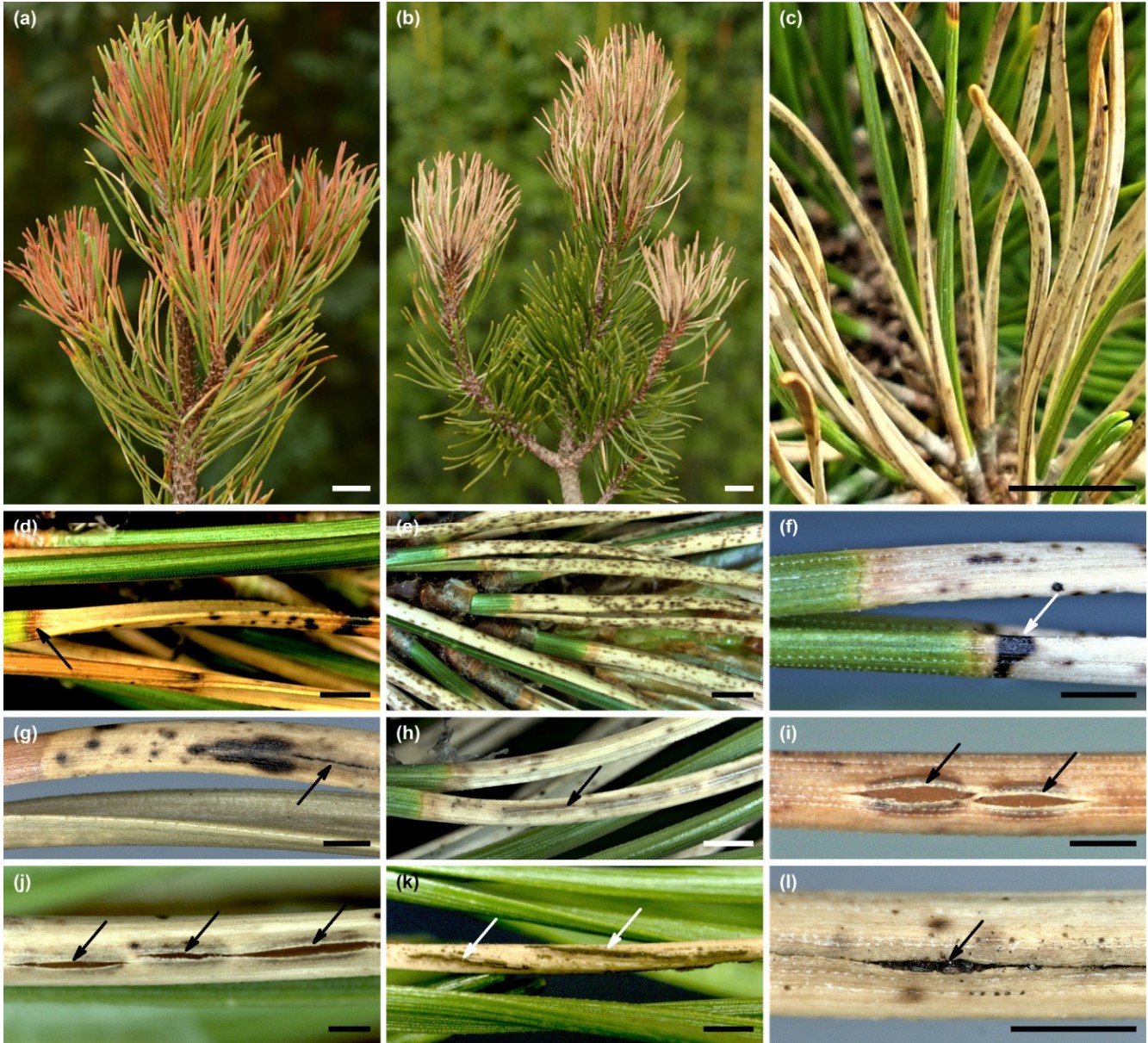

**Figure 2.** Symptoms on needles of *Pinus mugo* and etiological signs of *Lophodermella sulcigena* (**a**) reddish-brown current year needles (August 2017), (**b**) partly white-gray discolored previous year needles (May 2016), (**c**) hysterothecia on white-gray needles (July 2016), (**d**) red-brown band between living and dead parts of the needle (arrow), (**e**) numerous dark spots on dead distal parts of the needles, (**f**) black transverse stripe on dead part (arrow), (**g**) hysterothecium developed within dark spot (arrow), (**h**–**j**) hysterothecia: in dry condition (arrow) (**h**), in humid conditions (arrows) (**i**), after ejecting most of the ascospores (arrows) (**j**), (**k**,**l**) secondary fungi in/on hysterothecia: *Cladosporium herbarum* conidiophores and conidia (arrows) (**k**), *Epicoccum nigrum* sporodochia (arrow) (**l**). Scale bars: (**a**–**c**) = 1 cm, (**d**,**e**) = 0.2 cm, (**f**,**g**) = 0.1 cm, (**h**) = 0.2 cm, (**i**,**j**) = 0.1 cm, (**k**) = 0.2 cm, (**l**) = 0.1 cm.

**Table 2.** Decrease in radial growth of *Pinus mugo* shoots with diseased needles in 2016–2020.

| Years | 2016 | 2017 | 2018 | 2019 | 2020 |
|---|---|---|---|---|---|
| % of shoots showing a decrease in radial growth compared to the previous year | 42.9 | 57.1 | 65.7 | 62.9 | 56.7 |
| Average decrease in radial growth of shoots (%) | 22.1 | 20.0 | 22.2 | 24.7 | 25.8 |
| Minimal decrease in radial growth in a single shoot (%) | 4.1 | 2.3 | 3.8 | 2.7 | 4.6 |
| Maximal decrease in radial growth in a single shoot (%) | 68.1 | 57.1 | 65.7 | 72.6 | 57.9 |

*3.2. Morphology of Lophodermella sulcigena*

The basis for the identification of fungal species associated with disease symptoms on a morphological basis were hysterothecia (Figure 2h–j) and, to some extent, cultures isolated on agar medium (Figure 3a–c). Hysterothecia matured mainly in June–July on needles that had been infected the year before. They developed on both sides of the needles, on uniformly white-gray needles (Figure 2c) and those that showed numerous dark spots (Figure 2e–g). Hysterothecia were produced below the host hypodermis. They were elliptical or elongated in shape, 1.5–12 mm long, or longer if connected to each other, 0.3–0.5 mm wide and 0.16–0.3 mm deep. In mature ascocarps, a single longitudinal split appeared (Figure 2i,j), which is particularly well visible in the state of high humidity. Asci clavate, 110–140 × 12–15 μm (average 126.3 × 13.2 μm), 8 spored, rarely 4 spored (Figure 3d,e). Paraphyses filiform, unbranched, with numerous septa, 100–120 μm long, 2.0–3.0 μm wide, sometimes swollen at the tip, surrounded by a thin layer of mucus (Figure 3f). Ascospores clavate, tapering abruptly towards the base, straight or slightly curved, hyaline, often filled with granular content, 1-celled, 35–64 × 4.5–6.0 μm (average 45.5 × 4.7 μm), surrounded by a 2–4 μm thick gelatinous sheath, especially wider from both sides, much thinner on both ends of the spore (Figure 3g,h). Occasionally, single asci containing eight or six ascospores of different shapes and sizes were found within the hymenium. These were 1-celled, hyaline, subglobose, ellipsoidal, pyriform or clavate spores with dimensions of 16–33 × 8–10 μm (Figure 3i,j). Based on the morphological characteristics of the ascomata and dimensions of ascospores compared with literature (Table 3), the fungus causing the type of symptoms analyzed on *P. mugo* needles was identified as *Lophodermella sulcigena*. No other *Lophodermella* species were found during the research.

Out of 180 needle fragments obtained from needles with typical symptoms of the disease laid out on MEA in Petrie dishes, *L. sulcigena* colonies were obtained from only 20 (11.1%) fragments, mainly from current year's needles without ascomata from the border of the living and dead parts. The first colonies were obtained in November 2015 (Figure 3a–c). The colonies were mainly concentrated on the pine needle fragment, while they showed very limited growth on the MEA (Figure 3a,b). After a fragment of the medium with mycelium was transferred to the new Petrie dishes, the colony either did not grow at all or again the growth was very limited, and the medium became dark brown after about 10 weeks (Figure 3c). Colonies were gray-brown with a flocculent structure, adjacent to the medium towards the periphery, and the reverse was gray-pink (Figure 3b). Aerial hyphae were hyaline to olive-brown, 2.0 to 7.5 μm thick with circular or oval, intercalary or apical, thin-walled swollen cells 6–17.5 μm in diameter. In the substrate, numerous olive-brown swollen cells, irregular or circular, 8–20.0 μm in diameter or elongated, 15–25 × 10–12.5 μm were observed. Sexual or asexual morph in vitro were not seen.

PCR resulted in fragments of 1132 base pairs (bp) for the 18S, 1655 bp for the appended ITS-28S, and 827 bp for the TEF1 gene regions of the subset of isolates sequenced. The sequences of the TEF1 fragment included only an exon without introns. All sequences within the analyzed gene fragments showed 100% similarity.

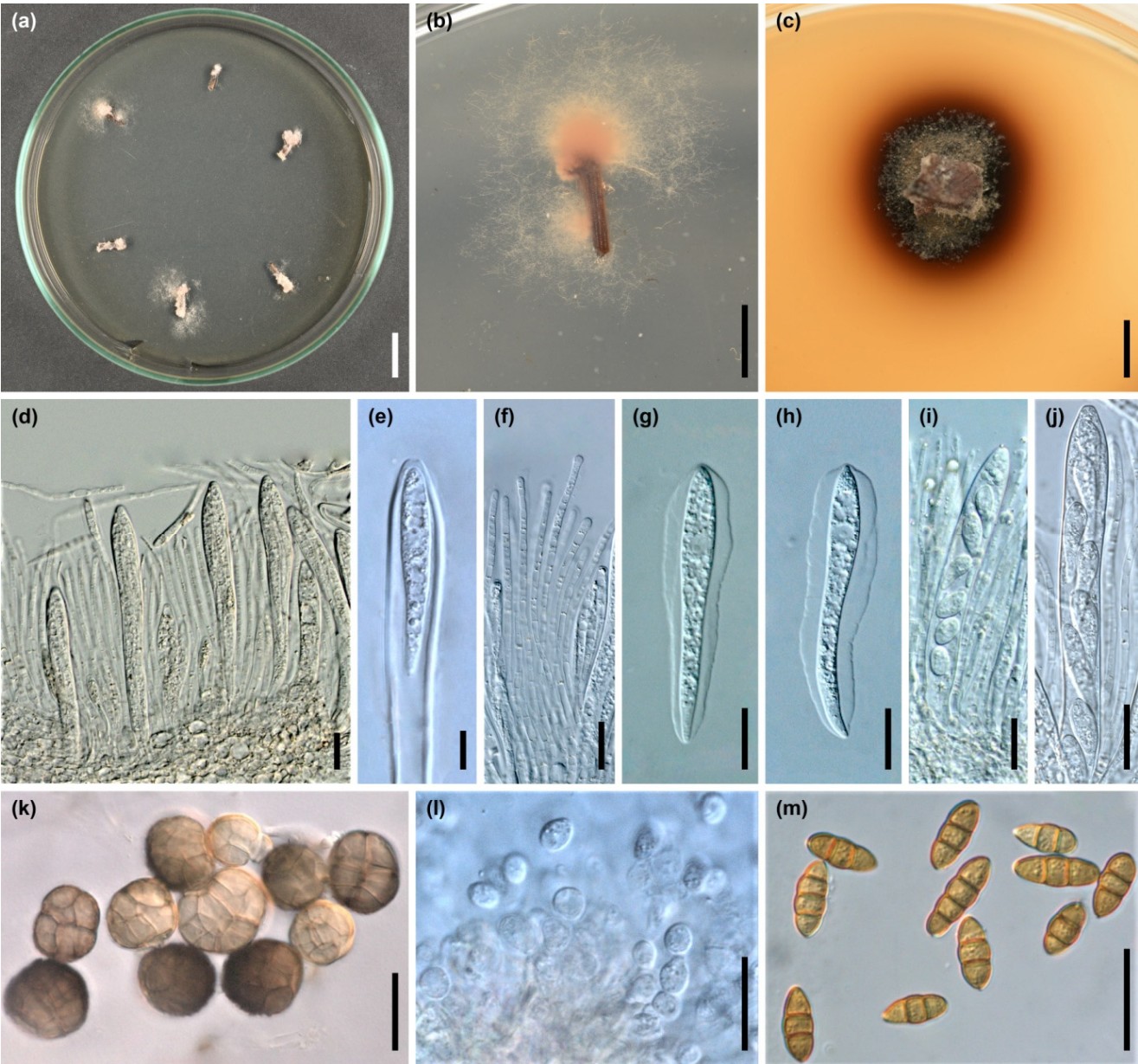

**Figure 3.** Colonies and microstructures of *Lophodermella sulcigena* and secondary fungi: (**a**) colonies growing from needle fragments (8 weeks, MEA, 20 °C), (**b**) reverse of two colonies, (**c**) 10-week old subculture on MEA, (**d–j**)—microstructures of *L. sulcigena*: fragment of hymenium with asci and paraphyses (**d**), ascus with one unejected ascospore (**e**), paraphyses (**f**), ascospores with gelatinous sheath (**g,h**), ascus with six unusually shaped ascospores (**i**), ascus with eight unusually shaped ascospores (**j**), (**k,l**)—secondary fungi in hysterothecia of *L. sulcigena*: *Epicoccum nigrum* conidia (**k**), *Seuratia millardetii* asexual stage, conidia and torulose cells in gelatinous matrix (**l**), *Hendersonia* sp. conidia (**m**). Scale bars: (**a–c**) = 1 cm, (**d**) = 20 μm, (**e**) = 10 μm, (**f**) = 20 μm, (**g,h**) = 10 μm, (**i–m**) = 20 μm.

The aligned data set for the concatenated ITS-28S-TEF1 gene region used in phylogenetic analysis included 29 taxa and 2190 characters (with gaps). The conducted phylogenetic analysis, based on the data available in GenBank for representatives of the Rhytismataceae family, showed the high usefulness of this method for identifying closely related species forming it. Members of the genus *Lophodermella* together with *Lophophacidium dooksii* form a clade with strong statistical support. The only exception is *Lophodermella conjuncta* forming a separate clade. The sequences of *L. sulcigena* isolates form a distinct group on the phylogenetic tree. *L. sulcigena* shows the greatest phylogenetic similarity to *Lophodermella*

*montivaga*, with which it forms a sister clade. The comparison of the DNA sequences of concatenated ITS-28S-TEF1 gene of selected isolates from Poland confirmed their high affinity to *L. sulcigena* PH18 0656 (Figure 4).

**Table 3.** Dimensions of ascospores given in the literature for *Lophodermella sulcigena* on *Pinus* species in Europe.

| Reference | Tree Species | Ascospores Length × Width (µm) |
| --- | --- | --- |
| Lagerberg (1910) [56] | *P. sylvestris* | 44–58 × 6 |
| Darker (1932) [40] | *P. mugo, P. sylvestris* | 27–35 × 4–5 |
| Terrier (1944) [57] | *P. mugo, P. sylvestris* | 35-(44.5–54)-65 × 4–6 |
| Moriondo (1963) [41] | *P. sylvestris* | 35–55 × 4–6 |
| Dennis (1968) [42] | *Pinus* spp. | 27–35 × 4–5 |
| Millar & Minter (1978) [43] | *P. mugo, P. nigra, P. sylvestris* | 27–40 × 4–5 |
| Kowalski (1988) [46] | *P. sylvestris* | 35–57 × 3.7–5 |
| Butin (2011) [14] | *P. mugo, P. nigra, P. sylvestris* | 27–35 × 4–5 |
| Beenken (2019) [58] | *P. mugo* | 35–65 × 4–6 |
| Kowalski et al. (2024) (Present study) | *P. mugo* | 35–64 × 4.5–6.0 |

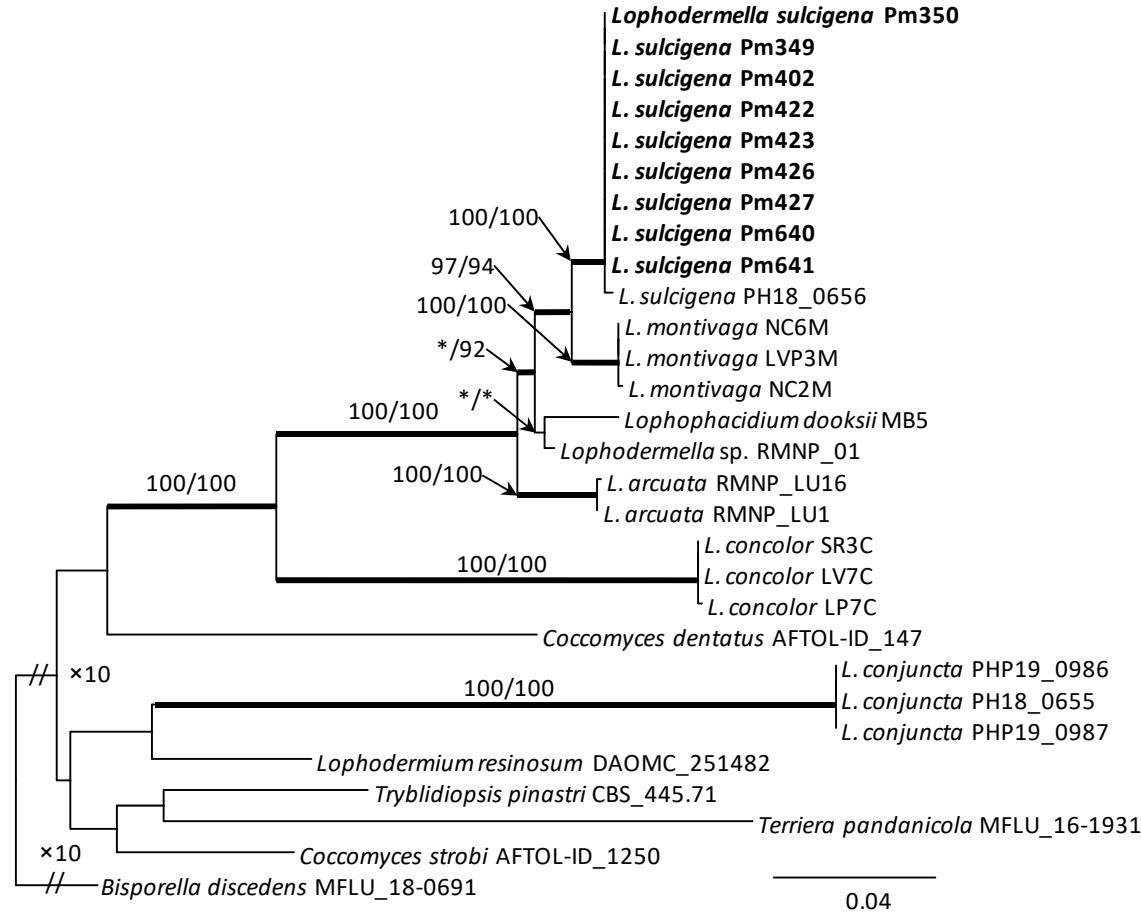

**Figure 4.** Phylogram from Maximum Likelihood (ML) analyses of the combined datasets of ITS + 28S + TEF1 for selected Rhytismataceae. Sequences obtained in this study are indicated with bold text. Bootstrap values ≥ 75% for ML and Maximum Parsimony (MP) analyses are presented at nodes as follows: ML/MP. Bold branches indicate posterior probability values ≥ 0.95 obtained from Bayesian Inference (BI) analyses. * Bootstrap values < 75%. The tree is drawn to scale (see bar) with branch lengths measured in the number of substitutions per site. *Bisporella discedens* was used as outgroup taxon.

*3.3. Secondary Fungi*

Fruit bodies of other fungi were also found on the needles of *P. mugo* primary killed by *L. sulcigena*; the more common of them are listed in Table 4. *Hendersonia* sp. were found at various locations on the needle, while *Leptomelanconium allescheri* and *Lophodermium* species were more often present in the apical part of the needle, and this zone was often separated by a brown fuzzy stripe. The second needle in the bundle was not infected by *L. sulcigena*, which underwent secondary death; no *Hendersonia* sp. was found, and apart from *Lophodermium* species, asexual morph of *Sydowia polyspora* appeared (Table 4). Various fungi were also found inside hysterothecia developed on killed needles (Table 4). Some of them were found already in the early stage of hysterothecium development, when numerous ascospores were still present in the asci (Table 4, Figure 3k–m). In the upper part of the hymenium, sporodochia with conidia (*Epicoccum nigrum*) (Figures 2l and 3k), stromatic colonies with torulose cells in a gelatinous matrix (asexual morph of *Seuratia millardetii*) (Figure 3l), or pycnidia with conidia (*Hendersonia* sp.) (Figure 3m) were identified. Some fungi were encountered in a later stage when ascospores from asci were mostly or all ejaculated. A cluster of their conidiophores and conidia could be observed outside the hysterothecia (Figure 2k).

**Table 4.** Fungi associated *Lophodermella sulcigena* on *Pinus mugo* needles.

| Substrate Type | Fungi [1] |
|---|---|
| Needle killed by *L. sulcigena* | *Hendersonia* sp.<br>*Leptomelanconium allescheri*<br>*Lophodermium conigenum*<br>*Lophodermium corconticum* |
| The second dead needle in the bundle not infected by *L. sulcigena* | *Lophodermium conigenum*<br>*Lophodermium corconticum*<br>*Sydowia polyspora* |
| Hysterothecium of *L. sulcigena* | *Alternaria* sp. (B)<br>*Cladosporium herbarum* (B)<br>*Cladosporium* sp. (B)<br>*Epicoccum nigrum* (A)<br>*Hendersonia* sp. (A)<br>*Seuratia millardetii* (A) |

[1] (A) early stage, second year of needles, summer; (B) late stage, second year of needles, autumn, and third year of needles.

## 4. Discussion

The results of this study have shown that a serious needle blight outbreak that occurred from 2015–2020 among *P. mugo* in high elevation forests of southern Poland was caused by *L. sulcigena*. Current analyses have not shown the occurrence of *L. conjuncta* in this area, despite the fact it was recorded both in the past [57] and recently on the needles of *P. mugo* in the Swiss Alps [58,59]. This species, unlike *L. sulcigena*, produces shorter hysterothecia and significantly longer ascospores [44,58]. Morphologically, *L. sulcigena* is very similar to *L. montivaga* found on pines in the USA. It was even speculated whether they were the same species [30,40,57]. However, recent molecular studies have shown that they are separate species [60]. Also, the current results indicate that *L. sulcigena* shows the greatest phylogenetic similarity to *L. montivaga*.

It was noticed relatively long ago that the most destructive of the *Lophodermella* species are those capable of completing their life cycles in one year, and *L. sulcigena* was included in this group [40]. *L. sulcigena* has so far been observed in Europe on two-needle pines: *P. mugo*, *P. nigra*, and *Pinus sylvestris* [41,44,57–59,61–65]. In Poland, this is the first observation of this pathogen on *P. mugo*. So far, *L. sulcigena* has been found in Poland only on *P. sylvestris* [46,47,66].

The currently observed symptoms on *P. mugo* were generally very similar to those of *L. sulcigena* on other pine species [41,44,62,64]. However, it has now been observed that

the infestation of needles by *L. sulcigena* was occasionally accompanied by an unusually high density of dark spots, which was not observed to such an extent in *P. sylvestris* needles [46,62]. According to Campbell [67], the small dark areas on the needle are the result of the accumulation of dark hyphae of *L. sulcigena* which develops later into the hysterothecium. Such dark spots were also found on needles affected by *L. conjuncta* [57].

Other authors [32] observed the transverse black bands at the edge of the fruiting areas on *P. contorta* infected by *L. montivaga*. Such symptoms have also been reported now, but sporadically. Needles infected by *L. sulcigena* take on a white-gray color. However, this is not unique to this pathogen. Such discoloration accompanies the infection of needles by *Leptomelanconium allescheri* as well [29]. Also secondary fungi, e.g., *Hendersonia acicola*, contribute to such a change in color of the needles [44,68]. Different reactions between various tree species are also observed in case of other plant pathogens. For example, needles infected by *Dothistroma septosporum* usually develop red bands, but depending on the host species and on the environment, in which the tree grows, may have a different color [69–71]. This is also reflected in different in vitro cultivation conditions [72].

It is generally known that *Lophodermella* does not grow well in agar culture, even with the addition of pine needle extract, and attempts to obtain pure cultures from single-ascospore isolations have failed [44,60]. Due to this fact, some authors classify them as fungi with potentially obligate lifestyle [60]. However, it was quite often possible to isolate *L. sulcigena* on agar medium from symptomatic needles of *P. sylvestris* [47,62]. The present study confirmed that it was possible to obtain cultures of *L. sulcigena* from *P. mugo* needles, although the growth of *L. sulcigena* on MEA was very limited. However, it should be concluded that the isolation of *L. sulcigena* on agar media cannot be the basis for the assessment of the frequency of this pathogen in needles.

Most of the morphological features of *L. sulcigena* on *P. mugo* in Polish Tatra were consistent with previous observations by other authors [41,43,44,58]. However, there are some differences in the length of ascospores. According to some authors, ascospores reach up to 35–40 μm in length [14,42,43], while according to other authors, ascospores reach up to 50–65 μm in length [41,46,56,58]. These differences may indicate that there are two subpopulations of this pathogen in Europe. However, it cannot be excluded that the different data are the result of analyses of the pathogen at different stages of development. In addition, the measurement results can be affected by the use of different mounting media and the measurement of dried but not living samples [73,74]. The present study also indicated that *L. sulcigena* hysterothecia may contain asci with eight or six ascospores of different shapes and sizes. This suggests that there may be some disturbance in the formation of spores. Terrier [57] stated that in some *L. sulcigena* asci instead of eight ascospores there were four because the other four degenerated. Instability of sporogenesis was also sometimes observed in other fungal species [75]. For example, in some *Pezicula* species, most asci contain eight spores, and some contain six or four spores. There are known cases that the four spores were not degenerated but showed spores smaller than usual. Verkley [75] considers that these changes may be influenced by adverse environmental conditions or some kind of genetic degeneration.

Possibilities of comparing *L. sulcigena* strains on a genetic basis are limited so far [60,76]. For the first time, one *L. sulcigena* sequence from *P. sylvestris* was deposited in 2021 [60]. These data have been enriched as a result of the present study with nine sequences from *P. mugo* growing in the Polish Tatra, obtained both on the basis of cultures and direct DNA isolation from symptomatic needles. Enlarging the collections of this type of data from different regions of Europe and different plant hosts should enable the analysis of the genetic diversity of this pathogen. Genetic analysis of another dangerous pathogen of *Pinus* species, *Gremmeniella abietina*, in Poland has shown that mountain populations differ from lowland populations [77].

Based on the results of the phylogenetic analysis, it can be concluded that the genus *Lophodermella* is not monophyletic. *Lophodermella conjuncta* individuals form a well-defined and externally located clade in relation to other species included in the genus *Lophodermella*.

A similar conclusion can be drawn from the work of Ata et al. [60]. This may indicate the need for changes in the taxonomy of species currently belonging to the genus *Lophodermella*.

Numerous fungal species associated with needles infected primary by *L. sulcigena* are known. On *P. sylvestris*, it is mainly *Hendersonia acicola* [44,47,68,78]. On *P. nigra*, heavy secondary colonization was shown by *Lophodermium seditiosum* and *Lo. conigenum* [44]. The currently observed colonization of *Lo. conigenum* and *Lo. corconticum* on *P. mugo* needles already infected by *L. sulcigena* may have been facilitated by the fact that both species occur in symptomless needles as endophytes [28]. In turn, *Leptomelanconium allescheri* is rather not dependent on the attack of other primary pathogens, as it can independently cause local necrotic lesion [29]. *Hendersonia acicola* and *H. pinicola*, through the secondary invasion of needles, may prevent the fructification of some *Lophodermella* species, which may contribute to the biological elimination of the disease [44,61,68,79]. On needles of *P. sylvestris* infected by *L. sulcigena*, the fungus *Epicoccum nigrum* was six times more common than in symptomless needles [47]. This fungal species, now found for the first time in *L. sulcigena* hysterothecia, is known for antifungal activities. It produces, i.al., flavipin, epicorazine A, epicorazine B, epipyrone and fungal cell wall degrading enzymes which take part in lysis of pathogen hyphae [80–82]. Also known among the *Cladosporium* genus are mycoparasites, as well as species producing various antibiotics with antifungal properties [83,84]. Another fungus, *Seuratia millardetii*, found presently in asexual morph, is a representative of the sooty mold that mostly grows superficially on plant tissue [45]. It can be assumed that the numerous gelatinous structures of this species found inside *L. sulcigena* hysterothecium may limit the spread of the pathogen's ascospores. Reducing the development of *L. sulcigena* by other fungi would be of great ecological importance, as *L. sulcigena* contributes to significant losses in attacked pine cultures [68]. Mitchell et al. [78] showed that the volume of diseased Corsican pine trees was reduced by 59% following a six-year epidemic. The reduction of the radial increment of *P. mugo* shoots was also confirmed in the present study. This was probably also influenced by other less frequent fungi on the needles of *P. mugo* [28,29].

## 5. Conclusions

For several years, a high intensity of *P. mugo* needle disease has been observed in one of the areas of its natural occurrence in the Tatra Mts. Previous preliminary studies showed that the mycobiota on diseased and dead needles of *P. mugo* is quite diverse. The current work was devoted to explaining the causal agent of the most common disease symptom on *P. mugo* needles. It was characterized by the fact that current year's needles died a few weeks after development, with their basal parts remaining green. The following year, numerous mature fungal fruit bodies (hysterothecia) appeared on such needles in early summer. Later, after the death of the basal parts, such needles gradually fell. Morphological analyses of microstructures of fungal fruit bodies produced on needles together with the use of molecular techniques showed that *Lophodermella sulcigena* is the cause of these symptoms. No other species of this genus were found on the examined needles. This is the first record of *L. sulcigena* on *P. mugo* in Poland. So far, it has only been found locally on *P. sylvestris*. Careful analysis of diseased needles was necessary for two main reasons: (a) symptoms on *P. mugo* needles were slightly different compared to *P. sylvestris*, especially regarding the intensity of dark spots, and (b) data on the dimensions of *L. sulcigena* ascospores are contradictory in the literature. The present study has provided further evidence of variation within *L. sulcigena* in terms of the presence of different numbers of spores in one ascus and the occurrence of ascospores of substantially different shapes and sizes. This is important data, as identification of this species can sometimes be problematic. In the course of the current research, cases of the presence of other, hitherto unknown fungi in the fruit body of *L. sulcigena* have been found. This creates an interesting direction of research on the possibilities of biological protection of *P. mugo* against *L. sulcigena*.

**Supplementary Materials:** The following supporting information can be downloaded at: https://www.mdpi.com/article/10.3390/f15030422/s1, Table S1: Characteristics of *Pinus mugo* needles with disease symptoms from previous year and current year, studied in 2016 in Chochołowska Valley (Tatra Mountains).

**Author Contributions:** Conceptualization, T.K., P.B. and C.B.; methodology, C.B. (field studies and sampling), T.K. (mycological aspects) and P.B. (molecular aspects); investigation, T.K., P.B. and C.B.; data curation, T.K. and P.B.; writing—original draft preparation, review and editing T.K. and P.B.; software, P.B.; visualization T.K. and P.B. All authors have read and agreed to the published version of the manuscript.

**Funding:** The research was partly financed by a subsidy from the forest fund transferred by the State Forests to the Tatra National Park in 2020.

**Institutional Review Board Statement:** Not applicable.

**Informed Consent Statement:** Not applicable.

**Data Availability Statement:** The data presented in this study are available in the NCBI GenBank database.

**Acknowledgments:** The authors thank T. Zwijacz-Kozica for enabling field research in the National Park.

**Conflicts of Interest:** The authors declare no conflicts of interest.

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
