# Peer review of "Involvement of Lophodermella sulcigena in Endemic Disease of Pinus mugo Needles in the Polish Tatra Mountains"

_forests, doi:10.3390/f15030422_

Round 1
Reviewer 1 Report
Comments and Suggestions for Authors
Dear authors,
Manuscript “Involvement of Lophodermella sulcigena in endemic disease of Pinus mugo needles in the Polish Tatra Mountains“ is precisely writtened. Abstract contains the most important parts of the manuscript which are appropriately described. Introduction appropriately describes importance of Pinus mugo and its diseases, with an emphasis on the study area, where serious symptoms of needle disease were observed. The aims of the study are stated. Methods are appropriate and described in detail, experimental design is suitable. Results are benefical and clearly presented. Number of samples is sufficient. Table and Figures contain many original data, which are clearly presented. Discussion contains appropriately compared results of the current study with relevant previous studies, and show suggestions for next studies. Conclusions are in accordance with the previous chapters. References are appropriate, actual and correctly cited in all parts of the manuscript, number of references (83) is sufficient.
The main aims were: “i) documentation of disease symptoms observed for the first time in Poland on P. mugo needles, and ii) identification and characterisation of the main causal agent on the morphological and molecular basis. An additional goal was to determine the phylogenetic position of identified causal agent in relation to closely related other species.” All these aims correspond to the study and were fulfilled. The fungus Lophodermella sulcigena were identified as the cause of the symptoms of the disease, and described in detail.
Lophodermella sulcigena was studied and described in previous studies in detail. However I consider this study is benefical, especially due to first confirmed occurrence of Lophodermella sulcigena on Pinus mugo in Poland, many data from several seasons, informations about fungi associated L. sulcigena on P. mugo needles, and valuable suggestions for further research.
Here are the specific comments:
Table 1: Symbol asterisk “*” is not explained.
Figure 2 and 3: Scale bar labels could be placed directly in the individual photos, e.g. above or under the scale bars. However, if there would be a problem with reading, the labels can be kept in the descriptions of the Figures.
Good luck with your research.
Reviewer.
Author Response
Dear Reviewers
We would like to thank the reviewers for their time and all comments regarding the manuscript, thanks to which its value was increased. Almost all remarks are included in the text. However, some aspects have not been investigated, so it is not possible to give a specific answer. Below we respond to individual remarks:
Reviewer 1
Thank you for your positive assessment of our research work.
Point 1: Table 1: Symbol asterisk “*” is not explained
Response 1: We explained this asterisk in Table1
Point 2: Figure 2 and 3: Scale bar labels could be placed directly in the individual photos, e.g. above or under the scale bars. However, if there would be a problem with reading, the labels can be kept in the descriptions of the Figures.
Response 2: We ask for your acceptance to leave the labels in the description of the Figures. Labels on the scale bar would cover important elements of symptoms or fungal structures in some Photos. For this reason, in Fig. 3 the scale bars are vertical and not horizontal.
Reviewer 2 Report
Comments and Suggestions for Authors
Overall, this manuscript provides a good summary of a serious needle blight outbreak within high elevation mugo pine forests in southern Poland. The results will be useful for future documentation in these forests and provide reference for needle blight outbreaks in similar, conifer-dominated forests at high elevations in other cold temperate regions of the world. My main question after reading this manuscript is what environmental stress predisposed or elicited the needle blight outbreak? Whenever a native pathogen causes such a significant disease outbreak, there is often some predisposing event that facilitated disease development. This is especially true for needle pathogens, which typically cause the most damage on drought-stressed and weakened trees. There was mention of the environmental stresses these trees face (L47-L55), but nothing specific about weather events preceding the outbreak. Was there an abundance of precipitation during the period of peak sporulation and spread and/or a significant drought that weakened the trees pre- or post-infection? It feels like there is more to this story that is not addressed here.
The English translation needs work throughout the manuscript. Presently, it cannot be published without a more thorough and accurate translation.
Some minor comments
Throughout manuscript: The authors routinely refer to the Tatra Mountains or Polish Tatra, but this is far too specific for a general reader that is not familiar with the region. The authors should describe the area more broadly at times, e.g. southern Poland or the Carpathian Mountains.
L99: More details on the plot size and location along the elevation gradient is required. Were there any differences in incidence and severity by site or elevation?
Table1: Remove the “Isolation Source” column since there is no variation among the isolates.
L81-82: Was disease severity rated in the study plots for comparison over the study period? It’s not clear when the outbreak first developed or ended (if it has ended).
Table 2: This table is large and not essential to the overall results. It should be moved to Supplemental Materials.
Table 4: Include ascospore dimensions from the current study for comparison as well.
Comments on the Quality of English Language
The authors should consult with a colleague that can assist with the English translation. For example, the start of the Discussion:
Current: “Studies have shown that the symptoms of a high intensity of the current year needles of P. mugo disease in the years 2015-2020 on the analyzed plots in the Polish Tatra should be attributed to L. sulcigena.”
Suggested: “The results of this study have shown that a serious needle blight outbreak that occurred from 2015-2020 on P. mugo in high elevation forests of southern Poland was caused by L. sulcigena.”
Author Response
Dear Reviewers
We would like to thank the reviewers for their time and all comments regarding the manuscript, thanks to which its value was increased. Almost all remarks are included in the text. However, some aspects have not been investigated, so it is not possible to give a specific answer. Below we respond to individual remarks:
Reviewer 2
Thank you for your positive assessment of our research work.
Point 1: My main question after reading this manuscript is what environmental stress predisposed or elicited the needle blight outbreak? Whenever a native pathogen causes such a significant disease outbreak, there is often some predisposing event that facilitated disease development There was mention of the environmental stresses these trees face (L47-L55), but nothing specific about weather events preceding the outbreak Was there an abundance of precipitation during the period of peak sporulation and spread and/or a significant drought that weakened the trees pre- or post-infection? It feels like there is more to this story that is not addressed here.
Response 1: The above aspects are very interesting and important for understanding the epidemiology of each disease. In high-mountain conditions, they are more difficult to determine than in lowland conditions, because the terrain is very variable, which results in the problem of varying snow coverage. To answer many questions, long-term monitoring of the health condition of trees and atmospheric factors would be necessary. We are currently unable to explain these problems. We started research in 2015 due to a disease outbreak. We have precisely defined the goals of the current work and adapted the methodology to achieve them.
Point 2: Throughout manuscript: The authors routinely refer to the Tatra Mountains or Polish Tatra, but this is far too specific for a general reader that is not familiar with the region. The authors should describe the area more broadly at times, e.g. southern Poland or the Carpathian Mountains.
Response 2: We provided this information in Abstract and in Introduction at the beginning of the text.
Point 3: L99: More details on the plot size and location along the elevation gradient is required. Were there any differences in incidence and severity by site or elevation?
Response 3: We completed the text.
Point 4: Table1: Remove the “Isolation Source” column since there is no variation among the isolates.
Response 4: We have removed the ‘Isolation Source’ column in Table 1
Point 5: 4 L81-82: Was disease severity rated in the study plots for comparison over the study period? It’s not clear when the outbreak first developed or ended (if it has ended).
Response 5: As we already reported in response to Point 1, there is no constant monitoring of the health condition of trees in this area and we do not know how the disease developed. We received permission from the Tatra National Park to conduct time-limited research. The scope of field observation to achieve the research objectives is given in section 2.1. Detailed quantitative data for 2016 are provided in Table 2 (currently Suppl. Table 1).
Point 6: Table 2: This table is large and not essential to the overall results. It should be moved to Supplemental Materials.
Response 6: As suggested, we have moved Table 2 to Supplemental Materials. This table contains important data, including the frequency of L. sulcigena in association with certain needle symptoms.
Point 7: Table 4: Include ascospore dimensions from the current study for comparison as well.
Response 7: As suggested, we have supplemented this data in Table 4 (currently Table 3)
Point 8: The authors should consult with a colleague that can assist with the English translation (An example is given).
Response 8: The text was read by a person proficient in English. Some changes have been made to the text.
Reviewer 3 Report
Comments and Suggestions for Authors
THE COMMENTS ARE PRESENTED IN THE ATTACHED PDF FILE

Author Response
Dear Reviewers
We would like to thank the reviewers for their time and all comments regarding the manuscript, thanks to which its value was increased. Almost all remarks are included in the text. However, some aspects have not been investigated, so it is not possible to give a specific answer. Below we respond to individual remarks:
Reviewer 3
Thank you for your positive assessment of our research work.
Point 1: L 55 The effect of these factors can be summarised
Response 1: It has been done, as suggested.
Point 2: L 97 Record the approximate incidence of the disease; give the approximate area of each plot
Response 2: This information has been completed
Point 3: L142 Was the anamorphic phase of some of these fungi or Lophodermella observed and identified
Response 3: This information has been completed. The anamorphic stage was not observed in Lophodermella. Among the fungi associated with Lophodermella, only Lophodermium spp. produced sexual and asexual morphs, the others only produced asexual stages.
Point 4: Line 152 Was the pathogenicity of pure strains proven by Koch postulates?
Response 4: In this study the pathogenicity of L. sulcigena was not tested. The use of pure strains of L. sulcigena for the test is limited because this species hardly (if at all) grows on agar medium (details in Discussion)
Point 5: To give references for the identification of Lophodermella species
Response 5: References have been added.